# Neurological Disorders of Patients Living with HIV Hospitalized in Infectious Departments of the Specialist Hospital in Lower Silesia in Poland

**DOI:** 10.3390/healthcare10081481

**Published:** 2022-08-07

**Authors:** Justyna Janocha-Litwin, Krzysztof Simon

**Affiliations:** 1Department of Infectious Diseases and Hepatology, Medical University Wroclaw, 50-367 Wroclaw, Poland; 2Department of Infectious Disease, Gromkowski Provincial Hospital, Koszarowa 5, 51-149 Wroclaw, Poland

**Keywords:** PLWH, Poland, neurological manifestation, neurotoxoplasmosis, neuroimaging

## Abstract

*Background and Objectives*: Central nervous system (CNS) disorders are estimated to occur in approximately 10–20% of people living with HIV (PLWH). They are more commonly observed in newly diagnosed patients and in previously untreated patients or those refusing to undergo antiretroviral treatment. CNS diseases can also be the first manifestation of HIV/AIDS infection. The most common HIV-related central nervous system diseases (CNS-D) are CNS toxoplasmosis, CNS cryptococcosis, progressive multifocal leukoencephalopathy (PML), and HIV-associated encephalopathy treated as a neurocognitive disorder. *Materials and Methods*: A retrospective analysis of available medical records was performed on 476 patients hospitalised over a period from 2016 to 2021 and diagnosed with HIV/AIDS infection at the department of infectious diseases at the Provincial Specialist Hospital in Wroclaw. An additional criterion for selecting patients for the analysis was the performance of head imaging using computed tomography or magnetic resonance imaging on prospective patients. *Results*: Neurotoxoplasmosis, neurocryptococcosis, progressive multifocal leukoencephalopathy (PML), and neurosyphilis were the most common CNS diseases among the analysed group of patients. Based on radiological descriptions, other abnormalities, such as vascular changes or cortical and subcortical atrophy of multifactorial origin, not exclusively related to HIV infection, were also frequently observed. The most common neurological symptoms reported in the study group were headaches, limb paresis, and gait and balance disturbance. *Conclusions*: The clinical picture and epidemiology of neurological manifestations in the group of HIV-infected patients under assessment were similar to the results of other authors. Given the current epidemiological situation, diagnosis for HIV infection should be considered in patients admitted to neurological departments.

## 1. Introduction

With the introduction of an effective combination anti-retroviral therapy (cART), the incidence of opportunistic diseases among patients with HIV (PLWH) has significantly decreased [1]. In a number of countries, including Poland, as many as 45–60% of patients are diagnosed with HIV/AIDS very late and at the stage of full-blown AIDS, usually with severe immunodeficiency and a CD4 T-lymphocyte count below 350 cells/mm^3^ (late presenters). This requires quick and complex diagnostics and the fastest possible introduction of cART therapy [2,3,4].

Neurological manifestations are observed in approximately 10–20% of patients with HIV, and they are among the most common complications in patients living with HIV. However, cases of neurological manifestations are observed in newly diagnosed or untreated patients, and they can very often be the first symptoms of HIV infection and/or full symptomatic AIDS. In European countries and in Poland, the most common AIDS-defining conditions affecting the central nervous system are CNS toxoplasmosis, CNS cryptococcosis, progressive multifocal leukoencephalopathy (PML), and HIV-associated encephalopathy treated as a neurocognitive disorder [5,6].

Among patients living with HIV, changes in the CNS most often relate to opportunistic diseases defining AIDS of infectious aetiology, i.e., toxoplasmosis, cryptococcosis, tuberculosis, CNS mycoses, progressive multifocal leukoencephalopathy, as well as AIDS-defining neoplasms, including brain lymphoma and Burkitt’s lymphoma. Moreover, in this group of patients, it is more likely to observe neurosyphilis or cortical and subcortical brain atrophy not associated with ageing, vascular changes or strokes. Often, other causes of CNS pathology unrelated to HIV infection coexist, such as post-traumatic lesions, neoplastic lesions other than lymphoma, and neuroinfections of different origin to those mentioned above [7]. The presence of HIV-specific lesions, such as HIV-associated neurocognitive disorder (HAND), vacuolar myelopathy, and peripheral neuropathy, should also be acknowledged; however, we did not take these into account while preparing this study.

Sadly, in European countries, despite good access to anti-retroviral therapy, laboratory diagnosis or CNS imaging, as well as the possibility of treatment for many opportunistic diseases, AIDS-defining conditions are still the direct cause of death among HIV-infected patients [8].

## 2. Objectives

The retrospective single-centre study aimed to assess the causes, prevalence, and clinical picture of CNS diseases in HIV-infected patients hospitalised in the Department of Infectious Diseases at the J. Gromkowski Provincial Specialist Hospital in Wrocław, Poland, over the period from 2016 to 2021.

## 3. Materials and Methods

A detailed retrospective analysis of medical records was performed on 476 patients hospitalised in our centre during the period from 2016 to 2021, diagnosed with HIV infection based on ICD-10 international classification codes: B20, B21, B22, B23, and Z21.

The main criterion for the inclusion of participants in our retrospective single-centre study was the neurological examination of the head during hospitalization using computed tomography or magnetic resonance imaging. The group of 200 patients selected on the basis of these 2 criteria (HIV infection and neuroimaging) were divided into two groups according to the presence of neurological symptoms. Subgroup A included patients with neurological symptoms that caused their admission to hospital or presented during hospitalization (121 patients), while subgroup B (79 patients) included patients who, on the basis of the available documentation and its retrospective analysis, did not report or present neurological abnormalities. The only exclusion criterion was incomplete documentation that did not allow selection of a patient for a particular subgroup.

The study design is presented in Figure 1.

The study was approved by the Bioethical Commission at the Medical University in Wroclaw and was conducted in accordance with the Declaration of Helsinki Ethical Principles.

This is one of the first studies of this type carried out on a Polish population.

### Statistical Analysis

The obtained results were analysed statistically. The values of the analysed quantitative parameters are presented by means of an average value; median, minimum and maximum values; lower and standard deviations; and qualitative parameters—count and percentage.

For qualitative parameters, the Chi^2^ test was used to assess the relationship between the distribution of the frequency of responses in the range of one variable. The normality of the distribution of variables in the study groups was checked using the Shapiro–Wilk normality test on the quantitative variables. Differences between the groups were assessed with Student’s *t*-test, and in the case of failure to meet the assumptions, with the U Mann–Whitney test. Differences between the three groups were assessed using the Kruskal–Wallis test. A significance level of *p* < 0.05 was adopted to indicate the existence of statistically significant differences or correlations. Database and statistical data were developed with the use of Statistica 9.1 software (StatSoft, Poland).

## 4. Results

In most cases, the reason for performing head imaging was the neurological symptoms, listed in Table 1 (subgroup A), in these patients. Most often, manifestations in the patients were chronic headaches, limb paresis, impaired consciousness, and balance and gait disturbances.

In subgroup B (79 patients without neurological symptoms), the reasons for neurological diagnosis combined with CNS imaging were:-Supplementation of the diagnosis of opportunistic diseases and patients with newly diagnosed HIV infection in the AIDS stage (34/79, 43.04%);-A follow-up after differential diagnosis in patients with symptoms other than neurological ones, e.g., patients with fever (25/79; 31.65%);-A follow-up diagnosis due to suspected neurosyphilis in patients with no symptoms due to syphilis of unknown duration or in cases of an insufficient decline in serum antibody titres after treatment (13/79; 16.45%) [9];-Head injury during hospitalisation (7/79; 8.86%).

As many as 67 out of the 200 patients, or 33.5%, were transferred from the neurology department to the department of infectious diseases, either because of abnormalities in patients already diagnosed with HIV or because the neurological symptoms and abnormalities discovered during imaging examinations were the reason for an additional follow-up diagnosis for HIV infection. The remaining patients were referred to the department of infectious diseases from the department of internal medicine (35/200; 17.5%); they visited the department themselves as walk-in patients (58/200; 29%), or they were patients of specialist clinics, the emergency department, or other departments (40/200; 20%).

Characteristics of the whole PLWH group and the subgroup of patients with neurological symptoms (A) and with no neurological symptoms (B), including selected demographic, epidemiological, and clinical data, are presented in Table 2, Table 3 and Table 4. In turn, the selected results of serological tests, including HBV infection (HBsAg only), anti-HBc total without determining the viral load of HBV DNA PCR, HCV (anti-HCV without determining the viral load of HCV RNA PCR), and syphilis (VDRL and specific tests, i.e., FTA, FTA-ABS, TPHA) for *T. gondii* infection are presented in Table 5.

There were no significant differences between subgroups A and B with regard to the duration of infection before hospital and neurological diagnosis, age of patients, CD4 T-cell count, death during hospitalisation, number of late presenters, results of anti-HCV tests, chronic and past HBV infection, history of toxoplasmosis, and HIV transmission route: IVDU, heterosexual or bisexual contacts or congenital infections.

There was a statistically significant difference between the study groups in terms of gender (*p* < 0.05), the possible transmission route among MSM (*p* < 0.01) and the results of serological tests (*p* < 0.01) or syphilis-specific tests (*p* < 0.05). There were statistically significantly more women in subgroup A (23.1%; 76.9% men) than in subgroup B (only 11.4% women and 88.6% men). In subgroup B, sexual transmission of HIV was statistically significantly more frequent (*p* < 0.01) among MSM (35.4%) than in subgroup A (17.4%). Positive results of screening tests for syphilis were noted in the subgroup of patients with no neurological disorders: 43.8% vs. 22.7% among patients with neurological symptoms. Similarly, positive results of specific tests in the corresponding subgroups (93.1% vs. 65.2%) were statistically significant. There was no statistically significant correlation between the other variables such as age or CD4 T-lymphocyte count.

Having analysed the entire study group of 200 patients (irrespective of the presence or absence of neurological symptoms), the most common CNS disorders were: toxoplasmosis, progressive multifocal leukoencephalopathy (PML), neurosyphilis, and cryptococcosis (Table 6). The above results are similar to the results obtained by other clinicians and data from the literature on the subject (see Discussion). In 10 patients, with a diagnosis identified as “other focal CNS signs”, the most common cause of CNS disorders were aspergilloma (1), metastatic tumours in the lungs (1), post-traumatic lesions (1), and ephedrone/permanganate encephalopathy (2). In five patients, the cause of CNS disorders could not be determined [10].

The comparison of subgroups A and B showed that in subgroup A, with statistically significant neurological symptoms, the most commonly identified were: cryptococcosis (*p* < 0.01; 11.6% vs. 0%), toxoplasmosis (*p* < 0.001; 20.6% vs. 2.5%), and PML (*p* < 0.05; 13.2% vs. 3.8%) (Table 6).

In turn, there was no statistically significant difference between subgroups with regard to the diagnosis of other CNS lesions, i.e., tuberculosis, other neuroinfections (of bacterial or viral aetiology), strokes, and neurosyphilis, or vascular changes or cortical and subcortical atrophy), although positive serology and confirmatory tests for syphilis were unexpectedly statistically significantly more frequent in subgroup B of patients with no neurological symptoms (Table 6 and Table 7).

Vascular changes as well as cortical and subcortical atrophy were not statistically significantly different in either subgroup (*p* = 0.14 and *p* = 0.09) (Table 7).

Having compared variables such as age, gender, CD4 T-lymphocyte count at the time of hospitalisation, number of years since diagnosis of the infection, transmission route, ARV treatment, and death during hospitalisation among patients with the most commonly diagnosed CNS opportunistic conditions, i.e., toxoplasmosis, cryptococcosis, and PML, there was no statistically significant differences with regard to those variables (Table 8 and Table 9). Two patients with coexisting toxoplasmosis and PML were not included in this analysis. A comparison of the three groups using the Kruskal–Wallis test (H) confirmed that there was no statistically significant difference in age (H = 5.18; *p* = 0.08), CD4 T-cell count (H = 5.66; *p* = 0.06), and the number of years since infection (H = 0.7; *p* = 0.70) (Table 5). A statistically significant correlation was noted between the transmission via intravenous drug use (*p* < 0.05) and the risk of death (*p* < 0.01) during hospitalisation among the assessed subgroups of patients diagnosed with cryptococcosis, toxoplasmosis, and PML (*p* < 0.05). The above-mentioned infection transmission route was more frequently observed among patients with cryptococcosis (57.1% of this group) and PML (47.1% of this group) than among patients with neurotoxoplasmosis (only 16% of this group), while death occurred in 58.8% of patients diagnosed with PML, 10% of patients with toxoplasmosis, and 14.3% with CNS cryptococcosis. With regard to other variables (Table 6), there were no statistically significant correlations: the distribution of gender (*p* = 0.05), late presenters (*p* = 0.58), and other infection transmission routes were similar in the groups under analysis. However, it should be emphasised that the assessment of such small groups with Pearson’s Chi-squared test may be unreliable.

It is worth noting that in all subgroups of patients, the median was patients with newly diagnosed HIV infections (we entered 0 for patients with a diagnosis made within a maximum of 1 week), those in the stage of a severe type of immunodeficiency (with the mean value and median of CD4 T-lymphocyte count below 100 cells/mm^3^), and those diagnosed with CNS opportunistic diseases (Table 8).

## 5. Discussion

We recognize the limitations of our single centre retrospective study in a small number of patients; moreover, not all participants of the study underwent the laboratory tests that we analysed, and it was not possible to obtain all epidemiological data from the available medical documentation.

In the assessed group of HIV-positive patients, the most frequently reported issues and symptoms indicative of CNS pathology were headaches, motor impairment in the limbs, and impaired consciousness; however, we did not take into account fever as a symptom not specific to neurological diseases, although it very often coexists in these diseases and cognitive disorders. The results are consistent with the outcomes presented by other authors; however, the fact that different symptoms have been selected for other studies should be emphasised. In the study by Hai Cehn et al., based on a very small group of PLWH, the following symptoms were most frequently described as those indicating a diagnosis of HIV/AIDS infection: limb paresis, neurocognitive disorders, and fever [11]; in the work by Indian authors: headaches and convulsions [12]; in Bolokadze et al.: headaches (91%), fever (75%), and focal defect symptoms (61%) [13]; and among Ethiopian patients: focal defect symptoms (65.4%), neurocognitive disorders (58.3%), and headaches (47.2%) [14,15].

Neurotoxoplasmosis, neurocryptococcosis, and PML were among the top diagnosed infections in HIV-positive patients with neurological symptoms: both in our work and in the study of a UK PLWH cohort [5]. In our previous paper, with regard to HIV-positive patients who died, we mentioned toxoplasmosis and cryptococcosis as the most common AIDS-defining conditions of CNS pathology [16]. However, it should be emphasised that HIV-related encephalopathy, recognised as one of the most common neurological diseases in this group of patients, was not included in our retrospective evaluation due to the lack of reliable data on that subject in the available medical documentation. Chinese scientists confirmed that the most common CNS disorders among PLWH are neurocryptococcosis, neurotoxoplasmosis, and CNS tuberculosis; the death rate in this group of patients is 13% [17]. However, in the group of Indian patients assessed by Shri Ram Sharma et al., the most common neurological manifestations were nervous system tuberculosis and cryptococcosis [12], which reflects the significant issue of mycobacterium tuberculosis infection in the Indian subcontinent. Similar results were presented by Georgian scientists who indicated CNS tuberculosis, neurotoxoplasmosis, and neurocryptococcosis as the predominant neurological manifestations, and these results are also presented by scientists from Morocco [14,15].

Most of the studies we refer to emphasise the positive correlation between the occurrence of CNS neurological diseases in HIV-positive patients with a stage of immunodeficiency characterised by a low CD4 T-lymphocyte count and an uncertain prognosis [18].

Attention should also be paid to cortical and subcortical atrophy and vascular changes in the brain described in brain imaging studies [19,20]. Sadly, due to the retrospective nature of this paper, we are not able to determine in what way the above-mentioned issues had a direct impact on the patients and whether they were of clinical significance, for example, in the diagnosis of HIV-related encephalopathy among those patients. Moreover, the formation of those abnormalities is influenced by, apart from the HIV virus, other factors, such as age; other viral infections, e.g., CMV, stimulants; and other chronic conditions. Additionally, these changes can coexist with each other as well as with other CNS conditions.

Therefore, we emphasise the need for a screening diagnosis for HIV infections among patients in neurological departments, and a follow-up neurological diagnosis with CT or MRI brain imaging in patients with diagnosed HIV infection in the stage of severe immunodeficiency and low CD4 T-lymphocyte count, including asymptomatic patients.

## 6. Conclusions

The most frequently presented neurological symptoms in the analysed group of HIV-positive patients were headaches, limb paresis, impaired consciousness, balance disturbances, and gait disorders.Cortical and subcortical atrophy, as well as vascular changes, are frequently observed pathologies among HIV-positive patients; however, their cause may be multifactorial and independent of HIV infection.Toxoplasmosis, progressive multifocal leukoencephalopathy, and cryptococcosis are among the most frequent CNS diseases observed in our patients with HIV/AIDS, irrespective of the presence or absence of neurological symptoms.The presence of CNS diseases, especially AIDS-related opportunistic diseases, is prognostically unfavourable.

## Figures and Tables

**Figure 1 healthcare-10-01481-f001:**
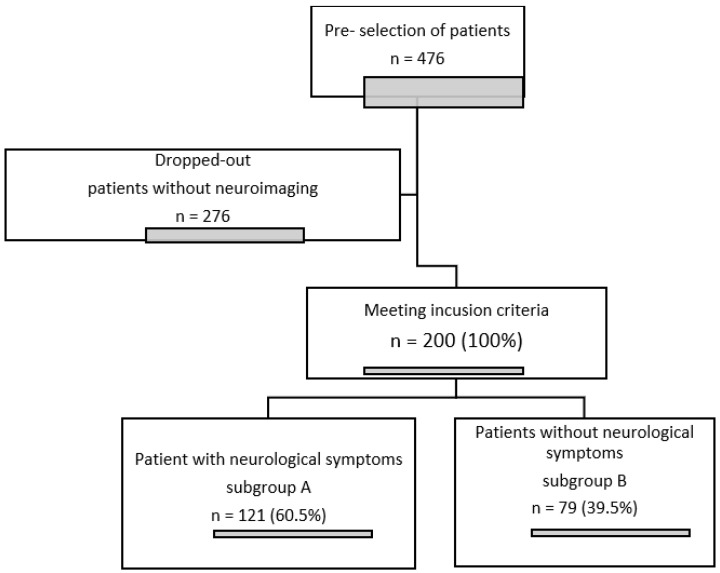
Flow chart of study design.

**Table 1 healthcare-10-01481-t001:** Neurological symptoms in the group of patients with neurological disorders (subgroup A).

Subgroup A	Number (*N* = 121)	% of Cases
Headaches	50	41.3%
Limb paresis	29	24.0%
Cranial nerve paresis	5	4.1%
Impaired consciousness	23	19.0%
Speech disorders	5	4.1%
Gait and balance disturbance/dizziness	24	19.8%
Vision disorders	15	12.4%
Epileptic seizures	14	11.6%
Behavioural disorders	18	14.9%

**Table 2 healthcare-10-01481-t002:** Age and time from detection of infection in individual patient subgroups.

	Entire Group *N* = 200			Subgroup with Neurological Symptoms (A) *N* = 121		Group with No Neurological Symptoms (B) *N* = 79		
	N	Average	Median Value	Average	Median Value	Average	Median Value	
Age	200	41.66	41	41.28	41	42.23	40	t = 0.673 *p* = 0.50
Current CD4 T-lymphocyte count	195	170	79	174.5	80	164.41	73	Z = 0.019 *p* = 0.984
Number of years since diagnosed HIV infection	200	3.7	0.5	3.72	0.5	3.68	1	Z = −0.292 *p* = 0.769

**Table 3 healthcare-10-01481-t003:** Sex distribution and the route of HIV infection in particular patient subgroups.

Assessed Variable		Entire Group N = 200		Subgroup A N = 121		Subgroup B	N = 79	Chi^2^ *p*
		N	%	N	%	N	%	
Gender	Female	37	18.5%	28	23.1%	9	11.4%	Chi^2^ = 4.375 ***p* < 0.05**
	Male	163	81.5%	93	76.9%	70	88.6%	
HIV transmission routes	Intravenous drug users (IVDU)	94	47.0%	63	52.1%	31	39.2%	Chi^2^ = 3.16 *p* = 0.08
	Men having sex with men (MSM)	49	24.5%	21	17.4%	28	35.4%	Chi^2^ = 8.45 ***p* < 0.01**
	Heterosexual contacts (HTX)	46	23.0%	29	24.0%	17	21.5%	Chi^2^ = 0.16 *p* = 0.69
	Bisexual contacts	3	1.5%	2	1.7%	1	1.3%	Chi^2^ = 0.14 *p* = 0.71
	Unknown	6	3.0%	4	3.3%	2	2.5%	Chi^2^ = 0.01 *p* = 0.91
	Congenital infection	3	1.5%	3	2.5%	0	0.0%	Chi^2^ = 0.66 *p* = 0.42

**Table 4 healthcare-10-01481-t004:** Death, late detection of HIV infection, and treatment of ARV in individual patient subgroups.

Assessed Variable		Entire Group N = 200		Subgroup A N = 121		Subgroup B N = 79		Chi^2^ *p*
		N	%	N	%	N	%	
Death		31	15.5%	22	18.2%	9	11.4%	Chi^2^ = 1.68 *p* = 0.19
Late presenters		104	52%	65	52.7%	39	49.4%	Chi^2^ = 0.36 *p* = 0.55
ARV treatment	de novo	77	38.5%	47	38.8%	30	38.0%	Chi^2^ = 0.05 *p* = 0.98
	YES	74	37%	45	37.2%	29	36.7%	
	NO	49	24.5%	29	24.0%	20	25.3%	

**Table 5 healthcare-10-01481-t005:** Serological studies in the field of viral hepatitis and syphilis in particular patient subgroups.

	Entire group N = 1P96		Subgroup A N = 117		Subgroup B N = 79		Entire group N = 156		Subgroup A N = 103		Subgroup B N = 53	
	Anti-HCV reactive						T. gondii IgG positive					
	N	%	N	%	N	%	N	%	N	%	N	%
YES	76	38.8%	46	39.3%	30	38.0%	79		57	55.3%	22	41.5%
NO	120	61.2%	71	60.7%	49	62.0%	77		46	44.7%	31	58.5%
Chi^2^ *p*	Chi^2^ = 0.36 *p* = 0.85						Chi^2^ = 2.68 *p* = 0.10					
	**Entire group** **N = 195**		**Subgroup A** **N = 79**		**Subgroup B** **N = 116**		**Entire group** **N = 195**		**Subgroup A** **N = 116**		**Subgroup B** **N = 79**	
	**HBsAg positive**						**Latent HBV infection** **(HBsAg negative, anti HBc total positive)**					
	**N**	**%**	**N**	**%**	**N**	**%**	**N**	**%**	**N**	**%**	**N**	**%**
YES	15	7.7%	11	9.5%	4	5.1%	58	29.7%	34	29.3%	24	30.4%
NO	180	92.3%	105	90.5%	75	94.9%	137	70.3%	82	70.7%	55	69.6%
Chi^2^ *p*	Chi^2^ = 0.75 *p* = 0.39						Chi^2^ = 0.26 *p* = 0.87					
	**Entire group** **N = 183**		**Subgroup A** **N = 110**		**Subgroup B** **N = 73**		**Entire group** **N = 52**		**Subgroup A** **N = 23**		**Subgroup B** **N = 29**	
	**VDRL positive**						**Positive syphilis test**					
	**N**	**%**	**N**	**%**	**N**	**%**	**N**	**%**	**N**	**%**	**N**	**%**
YES	57	31.2%	25	22.7%	32	43.8%	42	80.8 %	15	65.2%	27	93.1%
NO	126	68.8%	85	77.3%	41	56.2%	10	19.2%	8	34.8%	2	6.9%
Chi^2^ *p*	Chi^2^ = 9.12 ***p*** < **0.01**						Chi^2^ = 4.75 ***p*** < **0.05**					

**Table 6 healthcare-10-01481-t006:** CNS diseases in specific subgroups of HIV-infected patients.

CNS CHANGES	Entire Group N = 200		Subgroup A N = 121		Subgroup B N = 79		Chi^2^ *p*
	N	%	N	%	N	%	
CNS toxoplasmosis	27	13.5%	25	20.7%	2	2.5%	Chi^2^ = 13.45 ***p*** < **0.001**
CNS cryptococcosis	14	7%	14	11.6%	0	0%	Chi^2^ = 8.13 ***p*** < **0.01**
PML	19	9.5%	16	13.2%	3	3.8%	Chi^2^ = 3.9 ***p*** < **0.05**
CNS tuberculosis	3	1.5%	3	2.5%	0	0%	Chi^2^ = 0.66 *p* = 0.41
Neuroinfection	6	3.0%	6	5.0%	0	0%	Chi^2^ = 2.51 *p* = 0.11
Stroke	2	1.0%	2	1.7%	0	0%	Chi^2^ = 0.17 *p* = 0.67
Neurosyphilis	16	8.0%	8	6.6%	8	10.1%	Chi^2^ = 0.4 *p* = 0.53
Other	10	5.0%	6	5.0%	4	5.1%	Chi^2^ = 0.09 *p* = 0.77

**Table 7 healthcare-10-01481-t007:** Described vascular changes as well as cortical and subcortical atrophy in specific subgroups of HIV-positive patients.

	Entire Group N = 200		Subgroup A N = 121		Subgroup B N = 79		Chi^2^ *p*
	N	%	N	%	N	%	
Vascular changes	35	17.5%	25	20.6%	10	12.7%	Chi^2^ = 2.12 *p* = 0.14
Cortical and subcortical atrophy	22	11.0%	14	11.6%	8	10.1%	Chi^2^ = 0.08 *p* = 0.09

**Table 8 healthcare-10-01481-t008:** Epidemiological data and the number of CD4 T-cells in particular subgroups of patients with diagnosed CNS disease.

	CNS Cryptococcosis N = 14		Toxoplasmosis CNS N = 25		PML N = 17		H *p*
	Average	Median Value	Average	Median Value	Average	Median Value	
Age (number of years)	40.35	41	38.32	38	43.82	45	H = 5.19 *p* = 0.07
CD 4 T-lymphocyte count	30.64	22	84.9	47	100.53	27	H = 5.66 *p* = 0.70
How many years since the infection	3.28	0	2.1	0	2.9	0	H = 0.7 *p* = 0.70

**Table 9 healthcare-10-01481-t009:** Gender distribution, ARV treatment, and the route of HIV infection in particular subgroups of patients with diagnosed CNS disease.

		CNS Cryptococcosis N = 14		Toxoplasmosis CNS N = 25		PML N = 17		Chi^2^ *p*
		N	%	N	%	N	%	
Gender	Female	1	7.1	10	40%	3	17.7%	Chi^2^ = 5.87 *p* = 0.05
	Male	13	92.9%	15	60%	14	82.4%	
Late presenters		11	78.6%	21	84%	12	70.6%	Chi^2^ = 1.08 *p* = 0.58
Transmission routes	IVDU	8	57.1%	4	16%	8	47.1%	Chi^2^ = 7.98 ***p*** < **0.05**
	MSM	2	14.3%	8	32%	2	11.8%	Chi^2^ = 3.03 *p* = 0.22
	HTX	4	28.6%	11	44%	4	23.5%	Chi^2^ = 2.13 *p* = 0.34
	BI	0	0%	0	0%	1	5.9%	Chi^2^ = 2.34 *p* = 0.31
	Congenital	1	7.1%	1	4%	1	5.9%	Chi^2^ = 0.19 *p* = 0.91
	Unknown	0	0%	1	4%	1	5.9%	Chi^2^ = 0.8 *p* = 0.67
ARV treatment	de novo	6	42.9%	17	68%	9	52.9%	Chi^2^ = 5.09 *p* = 0.28
	YES	2	14.3%	5	20%	4	23.5%	
	NO	6	42.9%	3	12%	4	23.5%	
Death		2	14.3%	5	20%	10	58.8%	Chi^2^ = 9.49 ***p* < 0.01**

## Data Availability

Data supporting reported results can be provided upon request from the corresponding author.

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
