# Peer review of "Neurological Disorders of Patients Living with HIV Hospitalized in Infectious Departments of the Specialist Hospital in Lower Silesia in Poland"

_healthcare, 2022, doi:10.3390/healthcare10081481_

Round 1
Reviewer 1 Report
Dear Editor,
thank you for the opportunity to review this interesting manuscript. To be accepted this manuscript requires minor changes.
1. please consult native English speaker for minor language polishing (eg. it would be appropriate to use term disorders instead of disease in the abstract, gait and balance distubrance instead of gait disorders and balance disturance)
2. In the introduction section sentences: "The brain can be affected ... and "Knowledge of those abnormalities..." are not necessary - ant may be deleted.
3. In the Material and Methods section: please define clear inclusion and exclusion criteria. It is unclear why 276 patients were excluded (why they didn't require neuroimaging?. Please define groups in which the patients were divided. Numbers of patients which were included in a group are part of results. Please add if the study was approved by Ethic Committee.
4. In the results section please add a study flowchart and add details about study groups. It would be easier for readers to have baseline characteristics of patients. Generally, the results section is based on repeating the results from the tables. It would be more appropriate to focus on major findings.
5. In the Results section some references are missing (regarding positive correlation between occurrence of CNS neurological disorders in HIV patients)
6. The Conclusion section is consisted of several statements, it would be more appropriate to focus on major finding which is related to the study aim - as it is written in the Abstract.
Best regards,
Author Response
Dear Reviewer
Thank You very much for the positive review and very constructive comments, to which I will try to answer below, including them mostly in the corrections applied in the article.
- You're right - the article was sent for re-evaluation by the native translator
- I made corrections as suggested
3.The only inclusion criteria were:
-hospitalization in the infectious diseases ward
- HIV infection
- CT or MRI head imaging performed as part of hospitalization;
The analysis was performed retrospectively only on the basis of the available medical documentation; we did not evaluate the group of the remaining 276 patients from any angle; hence I cannot say whether these patients required neuroimaging or not.
The study, of course, was approved by the Ethics Committees.
Most likely, during the translation process, the sentence about diagnostic imaging was translated incorrectly.
The decision to select the group for analysis was made by the patients who underwent head imaging; not those who required neuroimaging as translated; the sentence has been corrupted
4, 5,6
Indeed, some of the results are repeated in the text; however, we wanted to point out that the results were statistically significant or interesting;
The conclusions were slightly revised in line with the comments
Unfortunately I have no experience in preparing a flowchart; hence I can not put it in the article
I added the statistical test results to table 2a.
The corrections were made in accordance with the recommendations of all reviewers, and I hope that the reasons for the article's final approval for publication
With gratitude

Reviewer 2 Report
The authors conducted a retrospective study by reviewing the medical record to assess CNS related diseases and behavioral characteristics among HIV patients who required a follow-up CNS diagnosis.
· Better aligning the introduction, objective, results and conclusions. Based on the first and second paragraph in the introduction, the authors indicated that CNS related symptoms should help for HIV diagnosis. The author stated that “it is always worth considering diagnosis for HIV infection among patients with neurological symptoms and/or with changes of unclear origin detected by CNS imaging”. Yet, the objective was to assess the causes, prevalence, and clinical pictures of CNS diseases among HIV-infected patients. In the method, the authors were looking at the CNS related symptoms among HIV patients who required a follow-up CNS diagnosis.
Better clarifying the hypothesis, analysis approaches taken in the method section to guide the readers. Also, Chi-square tests and normality tests are for quantitative data not qualitative. In addition, there were a good number of hypothesis testing were conducted. Multiple comparison adjustment will need to be considered.
I would also suggest the authors to provide some demographic characteristics information to help the readers to understand the sample.
Author Response
Dear Reviewer
Thank You very much for the positive review and very constructive comments, to which I will try to answer below, including them mostly in the corrections applied in the article.
As suggested, the sentence has been removed; contributed nothing to the analyzed assumptions.
After re-consulting with the statistician preparing the results; I was assured that the Chi-square test is applied to the qualitative variables that appear in the developed tables (2,3,4) with the results; while the Kruskal-Wallis test was used for the quantitative variables presented in Table 5.
Regarding the multiple comparison; statistics ensure that it should not be used in this case; as it is recommended when comparing more than 2 groups.
Tables 2a were supplemented with the results of the analysis with the t-test of the age of the patients; and the Mann-Whitney test for CD4 T cell count and time since HIV detection.
I added the statistical test results to table 2a and the entire article was read and revised by a native translator.
The corrections were made in accordance with the recommendations of all reviewers, and I hope that the reasons for the article's final approval for publication
With gratitude

Reviewer 3 Report
Obviously, the work is not original work, the results are not surprising, but are consistent with the results of many other works; however, I have not found any results from this part of Europe in the literature.I have not noticed any significant errors that caused the rejection of the article; hence, it makes a positive decision regarding the publication of the work at Helathcare
Please correction table 6- ivdu in capital letters- IVDU
Author Response
Dear Reviewer
Thank You very much for the positive review; the corrections were made in accordance with the recommendations of all reviewers, and I hope that the reasons for the article's final approval for publication
I added the statistical test results to table 2a and the entire article was read and revised by a native translator
With gratitude
Round 2
Reviewer 1 Report
Dear Editor, dear Authors,
thank you for the opportunity to review again the manuscript. Unfortunately, the authors didn't address all raised queries, and the manuscript still requires minor changes to be accepted. Please find my comments below:
1) Please clearly present inclusion and exclusion criteria for the study.
2) The second sentence of the second paragraph in the materials and methods section should be a part of the Results section - it might be the first sentence there. In the Materials and Methods section you may replace it with: The participants (or patients) were divided into two groups according to the presence of neurological symptoms. Please add in the Materials and Methods section that the study was approved by Ethic Committee
3) It would be appropriate to add flow chart which will show the patients selection and drop-outs (476 patients - 200 patients included, 276 excluded from the study - no neuroimaging - group with neurological symptoms (121) and group without neurological symptoms - (79)
4) The discussion section should be expanded with study limitation paragraph - single-center, retrospective, there are some laboratory exams that didn't perform on all patients, etc.
5) there are still some missing references in the discussion section (regarding the positive correlation between the occurrence of CNS neurological disorders in HIV+ patients.
6) The Conclusion section consists of several statements, it would be more appropriate to focus on major finding which is related to the study's aim - as it is written in the Abstract.
Best regards
Author Response
Dear Reviewer
Again, we are very impressed that you have read the article carefully and have made good comments to improve the quality of this article.
For each point, we have made the suggested corrections:
1) Please clearly present inclusion and exclusion criteria for the study.-ADDED
2) The second sentence of the second paragraph in the materials and methods section should be a part of the Results section - it might be the first sentence there. In the Materials and Methods section you may replace it with: The participants (or patients) were divided into two groups according to the presence of neurological symptoms. Please add in the Materials and Methods section that the study was approved by Ethic Committee- CORRECTED AND ADDED
3) It would be appropriate to add flow chart which will show the patients selection and drop-outs (476 patients - 200 patients included, 276 excluded from the study - no neuroimaging - group with neurological symptoms (121) and group without neurological symptoms - (79)-ADDED
4) The discussion section should be expanded with study limitation paragraph - single-center, retrospective, there are some laboratory exams that didn't perform on all patients, etc. -ADDED
5) there are still some missing references in the discussion section (regarding the positive correlation between the occurrence of CNS neurological disorders in HIV+ patients.- ADDE; HOWEVER, IF YOU HAVE YOUR OWN SUGGESTIONS, PLEASE INDICATE THEM SPECIFICALLY
6) The Conclusion section consists of several statements, it would be more appropriate to focus on major finding which is related to the study's aim - as it is written in the Abstract- THANK YOU HOWEVER, WE SUGGEST THAT YOU LEAVE THEM IN THE POINTS FOR CLARITY
Again, thank you for your insightful comments and observations, which will definitely improve the quality of the article
We ask for the final approval of the article if the changes are satisfactory.
Reviewer 2 Report
please see attached.

Author Response
As suggested, we propose to change the title to:
“Neurological disorders of patients living with HIV hospitalized in Infectious Departments of the Specialist Hospital in Lower Silesia in Poland”.
However, in-depth diagnostics among patients with neurological disorders takes place most often in hospital conditions, hence I do not think that the results would differ from other authors and publications
We did not find other similar Polish publications, we also believe that care in our country does not differ from medical care in other developed countries
The research and hypotheses were made on a small number of patients, but the statistical research was carried out with great reliability by cooperation with an experienced statistician
The cause of higher mortality in the group without neurological disorders may be other health problems among PLWH resulting also from the variety of health problems and the general condition of patients
We removed this conclusion from the article after the reviewers' first comments due to the inability to prove this hypothesis however the clinical picture and epidemiology of neurological manifestations in the our group of HIV-infected patients from Region of Lower Silesia under assessment were similar to
the results of observations of other authors. Given the current epidemiological situation, diagnosis
for HIV infection should be considered in patients admitted to neurological departments.
Again, thank you for your insightful comments and observations, which will definitely improve the quality of the article
We ask for the final approval of the article if the changes are satisfactory.
Reviewer 3 Report
The paper could be accepted without any further changes
Author Response
Thank You for the work put into the review of the article and its final approval :)
This manuscript is a resubmission of an earlier submission. The following is a list of the peer review reports and author responses from that submission.